# Synergistic Effects of Metal–Organic Nanoplatform and Guanine Quadruplex-Based CpG Oligodeoxynucleotides in Therapeutic Cancer Vaccines with Different Tumor Antigens

**DOI:** 10.3390/vaccines12060649

**Published:** 2024-06-11

**Authors:** Xia Li, Mitsuhiro Ebara, Naoto Shirahata, Tomohiko Yamazaki, Nobutaka Hanagata

**Affiliations:** 1Research Center for Macromolecules and Biomaterials, National Institute for Materials Science, 1-1 Namiki, Tsukuba 305-0044, Japan; 2Research Center for Materials Nanoarchitectonics (MANA), National Institute for Materials Science, 1-1 Namiki, Tsukuba 305-0044, Japan; 3Graduate School of Chemical Sciences and Engineering, Hokkaido University, Sapporo 060-8628, Japan

**Keywords:** TLR9 agonist, CpG oligodeoxynucleotides, nanoplatform, cancer vaccines, adjuvant

## Abstract

Oligodeoxynucleotides (ODNs) containing unmethylated cytosine–phosphate–guanosine (CpG) motifs are readily recognized by Toll-like receptor 9 on immune cells, trigger an immunomodulatory cascade, induce a Th1 -biased immune milieu, and have great potential as an adjuvant in cancer vaccines. In this study, a green one-step synthesis process was adopted to prepare an amino-rich metal–organic nanoplatform (FN). The synthesized FN nanoplatform can simultaneously and effectively load model tumor antigens (OVA)/autologous tumor antigens (dLLC) and immunostimulatory CpG ODNs with an unmodified PD backbone and a guanine quadruplex structure to obtain various cancer vaccines. The FN nanoplatform and immunostimulatory CpG ODNs generate synergistic effects to enhance the immunogenicity of different antigens and inhibit the growth of established and distant tumors in both the murine E.G7-OVA lymphoma model and the murine Lewis lung carcinoma model. In the E.G7-OVA lymphoma model, vaccination efficiently increases the CD4^+^, CD8^+^, and tetramer^+^CD8^+^ T cell populations in the spleens. In the Lewis lung carcinoma model, vaccination efficiently increases the CD3^+^CD4^+^ and CD3^+^CD8^+^ T cell populations in the spleens and CD3^+^CD8^+^, CD3^−^CD8^+^, and CD11b^+^CD80^+^ cell populations in the tumors, suggesting the alteration of tumor microenvironments from cold to hot tumors.

## 1. Introduction

The limited response rate of immune checkpoint therapies (10~40%) in clinical practice is renewing the call for cancer vaccines, which are believed to be the next frontier in cancer immunotherapy [1]. Considering the diversity and heterogeneity of tumors, which antigens to use and how to use immunostimulatory adjuvants will greatly influence the activation of antitumor immunity, the alteration of tumor immunosuppressive environments, and the efficacy of cancer vaccines [2,3,4,5,6,7,8].

In the 1890s, Dr. William Coley pioneered the injection of live bacteria into cancer patients to induce tumor regression, and later used heat-killed bacteria to reduce the lethality of infection, although their anti-tumor effects were confined to a subset of patients [9]. The mechanism of action of “Coley’s toxin” was initially attributed to endotoxin for decades, and it was not until the 1990s that the important role played by bacterial DNA in anti-tumor immunity was gradually recognized [9]. Oligodeoxynucleotides (ODNs) containing unmethylated cytosine–phosphate–guanosine (CpG) motifs, which are highly abundant in bacterial DNA, are readily recognized by Toll-like receptor 9 (TLR9) on immune cells, trigger an immunomodulatory cascade, and induce a Th1-biased immune milieu, which makes CpG ODNs have great potential to treat cancer, infections, and allergies [10].

Recent studies suggest that the physicochemical and biological properties of CpG ODNs, including resistance to nuclease degradation, half-life period, immunostimulatory effect and side effects, highly depend on their structure, nucleotide sequence, spatial distribution, the number of CpG motifs, and bases surrounding CpG motifs [11,12]. Because natural CpG ODNs with a phosphodiester (PD) backbone are susceptible to deoxyribonuclease (DNase) degradation, those with a phosphorothioate (PS) backbone have been developed to increase the resistance to nucleases [13], and have been utilized as adjuvants for the Hepatitis B vaccine since 2017, which is also the only FDA-approved use of CpG ODNs to date. However, repeated administration of PS-modified CpG ODNs is associated with adverse effects, such as lymphoid follicle destruction, reduced immune response, liver and renal damage, hemorrhagic ascites, and prolonged clotting [13,14,15]. The fabrication of an unmodified PD-backbone CpG ODNs with a higher-order structure is a promising approach to simultaneously enhance nuclease resistance and avoid the side effects of PS modification [13,16,17]. Moreover, due to their short half-life (less than 60 min), easy clearance from the body and low uptake efficiency by immune cells, the antitumor efficacy of free CpG ODNs monotherapy in clinical trials is limited and has not yet achieved clinical translation [13,18,19].

Nanoplatforms provide the possibility to prolong the retention of CpG ODNs inside the body, enhance the cellular uptake efficiency of CpG ODNs by immune cells, further protect CpG ODNs from nuclease degradation, and improve the therapeutic efficacy [13]. Especially, the optimization of nanoplatforms may realize the codelivery of immunostimulants and tumor antigens into the same antigen-presenting cells, reduce the non-specific immune stimulation, minimize side effects, and thus enhance the tumor antigen-specific immune response [4,5,20]. Metal–organic nanoplatforms built from metal ions/cluster nodes and organic ligands are promising for the codelivery of immunostimulants and tumor antigens due to their adjustable composition, porous structure, versatile functional groups, strong biomolecules loading ability, and biodegradability [9].

In this study, a green one-step synthesis process was adopted to prepare an amino-rich metal–organic nanoplatform (FN) using iron (III) chloride hexahydrate (FeCl_3_·6H_2_O) and 2-aminoterephthalic acid (NH_2_-C_6_H_3_-1,4-(COOH)_2_) in ethanol solvent at 70 °C for 4 h. The synthesized FN nanoplatform can simultaneously and effectively load model tumor antigens (OVA)/autologous tumor antigens (dLLC) and immunostimulatory CpG ODNs with an unmodified PD backbone and a guanine quadruplex structure to obtain various cancer vaccines. FN nanoplatforms and immunostimulatory CpG ODNs generate synergistic effects to enhance the immunogenicity of different antigens and inhibit the growth of established and distant tumors in both the mouse E.G7-OVA lymphoma model and the Lewis lung carcinoma model.

## 2. Materials and Methods

### 2.1. Preparation of CpG ODNs with Guanine Quadruplex Structure

Guanine quadruples forming CpG oligodeoxynucleotides (GD3) with a sequence 5‘-GGGTTGGGGTCGTTTTGTCGTTTTGTCGTTGGGTTGGG-3’, a length of 38 mer, a phosphodiester backbone, and an HPLC purification grade were custom-synthesized by Eurofins Genomics (Tokyo, Japan). Then, 2 mg/mL GD3 aqueous solution prepared using sterile Milli-Q water and Dulbecco’s phosphate-buffered saline (10 × D-PBS; nacalai tesque, Kyoto, Japan) were mixed at a volume ratio of 5:1, treated at 90 °C for 5 min, cooled to 30 °C for 60 min, and stored at 4 °C. The product obtained was a CpG ODN forming a guanine quadruplex structure, which was analyzed by circular dichroism (CD) spectroscopy [17].

### 2.2. Synthesis of Amino-Rich FN Nanoplatform

The synthesis of the amino-rich FN nanoplatform was performed from a solution of iron (III) chloride hexahydrate (40 mM) and 2-aminoterephthalic acid (20 mM) in ethanol solvent at 70 °C for 4 h. The obtained products were collected by centrifugation at 13,000 rpm for 10 min and washed by ethanol several times.

### 2.3. Preparation of Customized Cancer Vaccines Using ODNs, FN Nanoplatform and Tumor Antigens

In a typical synthesis, regarding cancer vaccines for E.G7-OVA lymphoma, the model tumor antigen OVA, ODNs, and FN nanoplatform were mixed in a final concentration of 1 μg/μL, 0.5 μg/μL, and 10 μg/μL, respectively, and then shaken for 1 h.

Regarding cancer vaccines for Lewis lung carcinoma, the synthesis process was performed as follows. Firstly, tumor cell lysate (dLLC) was prepared as autologous tumor antigen by repeatedly freezing and thawing 7 × 10^6^ cells/mL Lewis lung carcinoma cells (LLC, Bioresource Research Center, Tsukuba, Japan) in −30 °C and 4 °C, followed by the centrifugation at 1500 rpm for 3 min to obtain the supernatant. Then, autologous tumor antigen dLLC, ODNs, and FN nanoplatform were mixed in a final concentration of 0.06 μL/μL, 0.5 μg/μL, and 10 μg/μL, respectively, and then shaken for 1 h.

### 2.4. Physical and Chemical Characterization of Nanoplatforms and Cancer Vaccines

A high-resolution field emission scanning electron microscope (FE-SEM, Hitachi SU8000, Tokyo, Japan) with an energy-dispersive X-ray spectrometer (EDX) attachment was used to observe the morphology of the samples and analyze the elemental maps of cancer vaccines. A Fourier transform infrared spectrophotometer (IRTracer-100, Shimadzu, Kyoto, Japan) was employed to obtain the infrared absorption spectra of the samples. An ELSZ-1000Z analyzer (Otsuka Electronics, Hirakata-shi, Japan) was utilized to assess the zeta potentials of nanoplatforms and cancer vaccines. Dynamic light scattering analysis was performed using a DLS-8000HAL spectrophotometer (Otsuka Electronics, Japan).

A Micro BCA protein assay kit (Thermo Fisher Scientific, Waltham, MA, USA) was used to analyze the protein concentrations of model tumor antigen OVA or LLC tumor cell lysates before and after loading onto nanoplatform. A NanoDrop 2000 spectrophotometer (Thermo Fisher Scientific, USA) was used to analyze the concentration of ODNs. The loading efficiencies of protein or nucleic acid are calculated according to the following formula: loading efficiency = (initial concentration—final concentration after loading)/initial concentration × 100%.

### 2.5. In Vitro Assay

Bone marrow-derived dendritic cells (BMDCs) were isolated from C57BL/6J mice (5~6 weeks old, CLEA Inc., Tokyo, Japan) according to the following protocols. After the euthanasia of mice, bone marrow was harvested from the femurs and tibias. Then, erythrocytes lysis and the antibodies-mediated depletion of CD4^+^, CD8^+^, and I-A/I-E^+^ cells were carried out. The collected cells were subsequently cultured in RPMI 1640 medium containing 10% FBS and 20 ng/mL GM-CSF. On day 10, the loosely adherent and nonadherent cells were utilized as BMDCs.

To carry out the in vitro assay, BMDCs were seeded onto a cell culture plate at 2 × 10^5^ cells/well and then exposed to culture medium supplemented with different samples: (i) free OVA (final concentration: 5 μg/mL OVA); (ii) free OVA + ODNs (final concentration: 5 μg/mL OVA, 2.5 μg/mL ODNs); (iii) OVA + FN (final concentration: 5 μg/mL OVA, 25 μg/mL FN); and (iv) OVA + ODNs + FN (final concentration: 5 μg/mL OVA, 2.5 μg/mL ODNs, 25 μg/mL FN). One day or three days later, cytokine concentration in the supernatant was quantified using enzyme-linked immunosorbent assay (ELISA) kits (BD Biosciences, Franklin Lakes, NJ, USA).

### 2.6. In Vivo Antigen Cross-Presentation

Female C57BL/6J mice (5~6 weeks old, CLEA Inc.) were subcutaneously administered with 100 μL of the as-prepared samples: (i) saline; (ii) free OVA (100 μg/mouse); (iii) free OVA + ODNs (100 μg/mouse OVA, 50 μg/mouse ODNs); (iv) OVA + FN (100 μg/mouse OVA, 1 mg/mouse FN); and (v) OVA + ODNs + FN (100 μg/mouse OVA, 50 μg/mouse ODNs, 1 mg/mouse FN). About one day later, nearby draining lymph nodes were harvested from euthanized mice, ground, and passed through a 70 μm cell strainer to prepare single-cell suspensions. After pretreatment with anti-CD16/CD32 antibody to block the Fc receptor, the cells were stained with antibodies, such as PE-Cyanine7-anti-CD11c and APC-anti-H-2Kb of MHC class I bound to peptide SIINFEKL (BioLegend, San Diego, CA, USA). SP6800 spectral cell analyzer (Sony, Tokyo, Japan) was used for flow cytometry and FlowJo software was used to analyze the data.

### 2.7. In Vivo Anti-Tumor Experiments Using EG7-OVA

Thirty female C57BL/6J mice (5~6 weeks old, CLEA Inc.) were randomly divided into five groups. On day 0, mice were subcutaneously inoculated in the left flank with E.G7-OVA lymphoma cells (American Type Culture Collection, ATCC, Manassas, VA, USA) at an approximate dose of 1.8 × 10^5^ cells/mouse. On days 4, 7, and 10 after tumor inoculation, 100 μL of the following as-prepared samples were administered subcutaneously into the right flank of mice, respectively: (i) saline; (ii) free OVA (100 μg/mouse); (iii) free OVA + ODNs (100 μg/mouse OVA, 50 μg/mouse ODNs); (iv) OVA + FN (100 μg/mouse OVA, 1 mg/mouse FN); and (v) OVA + ODNs + FN (100 μg/mouse OVA, 50 μg/mouse ODNs, 1 mg/mouse FN). Tumor size measurement was performed using a digital caliper. The tumor is assumed to be an ellipsoid, and the volume can be calculated based on the following formula: π/6 × width × height × length. However, because height is difficult to measure, the following modified formula is widely used to calculate the tumor volume: 1/2 × tumor length × tumor width ^2^ [21].

At the endpoint of the anti-tumor experiment, spleens were harvested, ground, and passed through a 70 μm cell strainer to obtain single-cell suspensions. After pretreatment with anti-CD16/CD32 antibody to block Fc receptor, the cells were stained with FITC-anti-CD4, APC-Cyanine7-anti-CD8a (Biolegend), and APC-anti-T-Select H-2Kb OVA Tetramer-SIINFEKL (MBL) antibodies. SP6800 spectral cell analyzer (Sony, Japan) was used for flow cytometry and FlowJo software was used to analyze the data. In addition, cytokine content in the spleen was tested by ELISA kits (BD Biosciences) after being treated with a tissue protein extraction reagent (Thermo Fisher Scientific, USA).

### 2.8. In Vivo Anti-Tumor Experiments Using LLC

Twenty female C57BL/6J mice (5~6 weeks old, CLEA Inc.) were randomly divided into five groups. On day 0, mice were subcutaneously inoculated in the left flank with Lewis lung carcinoma cells at an approximate dose of 1.5 × 10^5^ cells/mouse. On days 4, 7, and 10 after tumor inoculation, 100 μL of the following as-prepared samples were administered subcutaneously into the right flank of mice, respectively: (i) saline; (ii) free dLLC (6 μL/mouse dLLC); (iii) free dLLC + ODNs (6 μL/mouse dLLC, 50 μg/mouse ODNs); (iv) dLLC + FN (6 μL/mouse dLLC, 1 mg/mouse FN); and (v) dLLC + ODNs + FN (6 μL/mouse dLLC, 50 μg/mouse ODNs, 1 mg/mouse FN). Tumor size measurement was performed using a digital caliper and tumor volume was calculated using the following formula: 1/2 × tumor length × tumor width^2^.

At the endpoint of the anti-tumor experiment, spleens and tumors were harvested, ground, and passed through a 70 μm cell strainer to obtain single-cell suspensions. After pretreatment with an anti-CD16/CD32 antibody to block the Fc receptor, the splenocytes were stained with PE-Cyanine7-anti-CD4, APC-anti-CD3, and APC-Cyanine7-anti-CD8a antibodies and the tumor cells were stained with FITC-anti-CD80, PE-anti-CD3, PE-Cyanine7-anti-CD11c, and APC-Cyanine7-anti-CD8a antibodies (BioLegend), respectively. SP6800 spectral cell analyzer (Sony, Japan) was used for flow cytometry and FlowJo software was used to analyze the data. In addition, cytokine content in the spleen was tested by ELISA kits.

### 2.9. Statistical Analysis

Statistical analysis was performed using a one-way analysis of variance (ANOVA) with Tukey’s multiple comparisons post hoc test. All the data are expressed in mean ± standard deviation (SD). Statistical significance is defined as a *p*-value less than or equal to 0.05.

## 3. Results

### 3.1. Characterization of FN Nanoplatform and Cancer Vaccines

The amino-rich FN nanoplatform was synthesized using iron (III) chloride hexahydrate (40 mM) and 2-aminoterephthalic acid (20 mM) in ethanol solvent at 70 °C for 4 h (Figure 1a). The synthesized particles exhibit a spindle-like morphology with lengths of approximately 200 nm and widths of about 100 nm (Figure 1b), which is the typical morphology of MIL-101 (Fe). Fourier transform infrared spectroscopy (FTIR) was used to analyze the reactant 2-aminoterephthalic acid and the reaction product FN (Figure 1c). In the FTIR spectrum of 2-aminoterephthalic acid, a C=O stretching band at 1668 cm^−1^ is observed due to its carboxylic acid group. Meanwhile, in the FTIR spectrum of the reaction product, the corresponding absorption band is shifted to 1572 cm^−1^, indicating the formation of iron (III) aminoterephthalate. In addition, the reaction product also has N-H stretching bands in the range of 3300~3500 cm^−1^ and a C-N stretching band near 1250 cm^−1^, due to an amino group of iron (III) aminoterephthalate.

The amino-rich FN nanoplatform was mixed with model tumor antigen OVA and CpG ODNs with a guanine quadruplex structure to prepare cancer vaccines for E.G7-OVA lymphoma (OVA + ODNs + FN). The resulting cancer vaccines maintain the spindle-like morphology of the FN nanoplatform, and energy dispersive X-ray (EDX) mapping analysis suggests that the sulfide (S)-containing OVA model antigen and phosphorus (P)-containing ODNs are homogeneously adsorbed into the FN nanoplatform (Figure 1d). The zeta potentials of the FN nanoplatform and E.G7-OVA lymphoma cancer vaccines are centered at 40 and 20 mV, respectively (Figure 1e). The adsorption and loading of the negatively charged OVA and ODNs molecules into the positively charged FN nanoplatform should account for the difference in zeta potentials before and after loading. The hydrodynamic particle sizes of the FN nanoplatform and E.G7-OVA lymphoma cancer vaccines are approximately 200 nm and 1.2 μm, respectively, as determined by dynamic light scattering analysis (Figure 1f). The partial aggregation of particles during the loading process results in the increase in the hydrodynamic particle sizes after loading. The concentration of protein and nucleic acid before and after loading was analyzed using a Micro BCA protein assay kit and a NanoDrop 2000 spectrophotometer, respectively. The loading efficiencies of OVA and ODNs for E.G7-OVA lymphoma cancer vaccines are about 96% and 86%, respectively (Figure 1g). Moreover, cancer vaccines for Lewis lung carcinoma were prepared by mixing the FN nanoplatform, dLLC tumor cell lysate and ODNs. The loading efficiencies of dLLC tumor cell lysate and ODNs are about 78% and 86%, respectively (Figure 1h).

### 3.2. In Vitro Evaluation of BMDC Activation

Primary BMDCs from mice were used to investigate the ability of the prepared cancer vaccines to activate the antigen-presenting cells (APCs), and the cytokine secretion was measured using an ELISA assay (Figure 2). Here, the OVA + ODNs + FN group stimulates BMDCs to secrete interleukin 12 (IL-12) more strongly than the other groups, such as medium, free OVA, free OVA + ODNs, and OVA + FN. Moreover, the OVA + ODNs + FN group and the OVA + FN group exhibit higher tumor necrosis factor—α (TNF-α) secretion than the other groups, such as medium, free OVA, and free OVA + ODNs.

### 3.3. In Vivo Antigen Cross-Presentation

To study the cross-presentation of the OVA model tumor antigen *in vivo*, the prepared samples were injected subcutaneously into the flank of C57BL/6J mice. About one day later, the draining lymph nodes were harvested, and flow cytometry was used to analyze the percentage of MHC I^+^ in CD11c^+^ cell population as an indicator of OVA antigen cross-presentation. In this study, the OVA + ODNs + FN group shows the highest average level of MHC I^+^ in CD11c^+^ cell population among all the groups (Figure 3). Furthermore, the OVA + ODNs + FN group shows a significantly higher level of MHC I^+^ in CD11c^+^ cell population than the free OVA + ODNs group, indicating the important role of the FN platform in enhancing the antigen cross-presentation.

### 3.4. In Vivo Anti-Tumor Efficacy in Therapeutic E.G7-OVA Lymphoma Mouse Model

Thirty C57BL/6J mice were randomly divided into five groups, and subcutaneously inoculated with E.G7-OVA lymphoma cells (1.8 × 10^5^ cells/mouse) in the left flank. Subcutaneous administration of the as-prepared cancer vaccines (OVA + ODNs + FN) into the right flank was then carried out on days 4, 7, and 10 after tumor inoculation. Moreover, saline, free OVA, free OVA + ODNs, and OVA + FN were administered as controls. The size of the tumor was monitored to study the effects of the different samples (Figure 4). Comparing the saline and free OVA groups, we can infer that antigen alone has little effect on inhibiting tumor growth. The addition of ODNs or FN to tumor antigen OVA can partially inhibit the growth of a distant tumor compared with the free OVA group. The OVA + ODNs + FN group shows the most effective inhibition of distant tumor growth among all the groups, suggesting that the simultaneous addition of ODNs and FN to the tumor antigen OVA can generate synergistic effects.

To understand the underlying antitumor mechanism, the spleens of mice at the endpoint were harvested to analyze CD4^+^, CD8^+^, and tetramer^+^CD8^+^ T cell populations (Figure 5). The average levels of the CD4^+^ T cell population in splenocytes of the saline, free OVA, free OVA + ODNs, and OVA + ODNs + FN groups are 16.17%, 18.07%, 16.57%, and 21.20%, respectively. The CD4^+^ T cell population in splenocytes of the OVA + ODNs + FN group is significantly higher than those of the other three groups, including the saline, free OVA, and free OVA + ODNs groups. The average levels of the CD8^+^ T cell population in splenocytes of the saline, free OVA, free OVA + ODNs, and OVA + ODNs + FN groups are 10.57%, 10.63%, 11.00%, and 12.17%, respectively. The CD8^+^ T cell population in splenocytes of the OVA + ODNs + FN group is significantly higher than those of the saline and free OVA groups. The tetramer^+^CD8^+^ T cell populations in the OVA + ODNs and OVA + ODNs + FN groups are significantly higher than those of the saline groups. The average levels of cytokines such as IFN-γ, IL-12, and TNF-α in the spleens of the OVA + ODNs + FN group are the highest among all the groups.

### 3.5. In Vivo Anti-Tumor Efficacy in Therapeutic Lewis Lung Carcinoma Mouse Model

Twenty C57BL/6J mice were randomly divided into five groups and were subcutaneously injected with Lewis lung carcinoma cells in the left flank (Figure 6). Subcutaneous administration of the as-prepared cancer vaccines (dLLC + ODNs + FN) into the right flank was then carried out on days 4, 7, and 10 after tumor inoculation. Moreover, mice administered with saline, free dLLC, free dLLC + ODNs, and dLLC + FN were used as controls. The tumor growth curve of the free dLLC group is close to that of the saline group, suggesting that only dLLC autologous tumor antigen fails to inhibit the tumor growth. Both the free dLLC + ODNs and dLLC + FN groups show a smaller tumor volume than the saline and free dLLC groups, which indicates that ODNs or FN can enhance the immunogenicity of dLLC autologous tumor antigen. More importantly, the dLLC + ODNs + FN group demonstrates the most effective inhibition of distant tumor growth among all the groups, which shows the same tendency as the therapeutic mouse E.G7-OVA lymphoma model.

To investigate the underlying antitumor mechanism, the CD3^+^CD4^+^ and CD3^+^CD8^+^ T cell populations in the spleens were quantified at the endpoint of the antitumor experiments (Figure 7). The average levels of the CD3^+^CD4^+^ T cell population in splenocytes of the saline, free dLLC, free dLLC + ODNs, dLLC + FN, and dLLC + ODNs + FN groups are 17.33%, 16.65%, 18.33%, 20.50%, and 22.23%, respectively. The CD3^+^CD4^+^ T cell population in splenocytes of the dLLC + ODNs + FN group is significantly higher than those of the saline and free dLLC groups. The average levels of the CD3^+^CD8^+^ T cell population in splenocytes of the saline, free dLLC, free dLLC + ODNs, dLLC + FN, and dLLC + ODNs + FN groups are 11.35%, 10.55%, 12.55%, 13.25%, and 15.10%, respectively. The CD3^+^CD8^+^ T cell population in splenocytes of the dLLC + ODNs + FN group is significantly higher than those of the saline, free dLLC, and free dLLC + ODNs groups. The average levels of cytokines such as IFN-γ, IL-12, and TNF-α in the spleens of the dLLC + ODNs + FN group are higher than those of the other groups.

To further study the underlying antitumor mechanism, CD3^+^CD8^+^, CD3^−^CD8^+^, and CD11b^+^CD80^+^ cell populations in the tumors were quantified at the endpoint of the antitumor experiments (Figure 8). The average levels of the CD3^+^CD8^+^ T cell population in tumors of the free dLLC, free dLLC + ODNs, dLLC + FN, and dLLC + ODNs + FN groups are 5.17%, 7.83%, 9.38%, and 10.47%, respectively. The CD3^+^CD8^+^ T cell populations in tumors of the dLLC + FN and dLLC + ODNs + FN groups are significantly higher than those of the free dLLC and free dLLC + ODNs groups. The average levels of the CD3^−^CD8^+^ cell population in tumors of the free dLLC, free dLLC + ODNs, dLLC + FN, and dLLC + ODNs + FN groups are 4.44%, 10.93%, 4.00%, and 9.69%, respectively. The CD3^−^CD8^+^ T cell populations in tumors of the free dLLC + ODNs and dLLC + ODNs + FN groups are significantly higher than those of the free dLLC and dLLC + FN groups. The average levels of the CD11b^+^CD80^+^ cell population in tumors of the free dLLC, free dLLC + ODNs, dLLC + FN, and dLLC + ODNs + FN groups are 2.73%, 2.40%, 5.97%, and 4.71%, respectively. The CD11b^+^CD80^+^ T cell populations in tumors of the dLLC + FN and dLLC + ODNs + FN groups are significantly higher than those of the free dLLC and free dLLC + ODNs groups.

## 4. Discussion

Metal–organic nanoplatforms are promising in drug delivery systems and cancer treatment due to their adjustable composition, porous structure, versatile functional groups, strong biomolecules loading ability, and biodegradability. However, metal–organic nanoplatforms are generally synthesized using toxic solvent, such as N,N-dimethylformamide, through a solvothermal method over 100 °C for 1~2 days. Regarding their biomedical application, there is great concern about the toxicity of residues. In this study, an amino-rich FN nanoplatform was synthesized using ethanal solvent at 70 °C for only 4 h. The use of ethanol green solvent with high biocompatibility is beneficial for its biomedical application. The synthesized amino-rich FN nanoplatform can simultaneously and effectively load model tumor antigens (OVA)/autologous tumor antigens (dLLC) and immunostimulatory CpG ODNs via electrostatic and aromatic interactions to obtain customized cancer vaccines.

CpG ODNs are readily recognized by TLR9 on immune cells and trigger an immunomodulatory cascade that involves antigen-presenting cells, T cells, and natural killer cells, which makes them promising for cancer immunotherapy. Because natural PD-backbone CpG ODNs are easily degraded by DNase, those with a PS backbone with enhanced stability [13] were approved as adjuvants for the Hepatitis B vaccine in 2017. However, repeated administration of PS-modified CpG ODNs resulted in several side effects, such as lymphoid follicle destruction, reduced immune response, liver and renal damage, hemorrhagic ascites, and prolonged clotting [13,14,15]. In this study, unmodified PD-backbone CpG ODNs with a higher-order guanine quadruplex structure and high nuclease resistance [17] were firstly used to evaluate their antitumor efficacy. Moreover, the amino-rich FN nanoplatform was used to co-deliver CpG ODNs and tumor antigens to further enhance their immunostimulatory effect and antitumor efficacy.

When coculture with primary BMDCs are in vitro, the combination of an amino-rich FN nanoplatform and CpG ODNs more effectively stimulates the secretion of Th1 cytokines such as IL-12 and TNF-α. IL-12 is a proinflammatory cytokine secreted by antigen-presenting cells such as dendritic cells and macrophages [22]. IL-12 is known to promote the differentiation of Th0 lymphocytes into a Th1 phenotype, enhance the cytolytic activities of cytotoxic T lymphocytes and natural killer (NK) cells, and facilitate the transformation of the immunosuppressive tumors into immunologically active tumors [22]. TNF-α is known to be an effective tumoricidal cytokine for inducing the apoptotic cell death of tumor cells and inhibiting the tumor growth in a dose-dependent manner [22]. Herein, the combination of an amino-rich FN nanoplatform and CpG ODNs effectively enhances the secretion of Th1 cytokines by antigen-presenting cells.

In this study, two different tumor-bearing mouse models, e.g., the E.G7-OVA lymphoma model and the Lewis lung carcinoma model, were used to investigate the antitumor efficacy of an amino-rich FN nanoplatform loaded with different tumor antigens and immunostimulatory CpG ODNs. Here, we confirmed the synergistic antitumor effects of an amino-rich FN nanoplatform and immunostimulatory CpG ODNs using these two completely different types of tumor models. This preparation method can be generalized to other tumors by using different tumor antigens. In addition, it should be mentioned that an FN nanoplatform loaded with CpG ODNs is designed to be the adjuvant for therapeutic cancer vaccines in this study. Therefore, subcutaneous administration of the as-prepared cancer vaccines was employed to trigger a systemic antitumor immune response, and the inhibition effects on the distant tumors were investigated. In fact, intratumoral or intravenous administration of an FN nanoplatform loaded with CpG ODNs as a medicine is also promising for cancer treatment, and the relevant work using different antitumor experimental processes will be considered in the future.

For mouse models and different groups used in this study, the CD4^+^ and CD3^+^CD4^+^ cell populations in the spleen are almost the same, and the CD8^+^ and CD3^+^CD8^+^ cell populations in the spleen are almost the same, as shown in Figure 7. But for tumor tissue, the situation is much more complicated. The CD3^+^CD8^+^ cell populations in tumors represent the infiltration of cytotoxic T cells in the tumor sites, while the CD3^−^CD8^+^ cell populations can represent NK cells in the tumor sites in this study [23]. Therefore, we stained the cells derived from the spleen and tumor in the Lewis lung carcinoma-bearing mouse model using an anti-CD3 antibody to clarify this difference in splenocytes and tumor cells.

In both the E.G7-OVA lymphoma model and the Lewis lung carcinoma model, vaccination using an amino-rich FN nanoplatform loaded with tumor antigens and immunostimulatory CpG ODNs greatly enhances the cell populations of typical T lymphocytes in splenocytes such as CD4^+^ and CD8^+^ T cells. Traditionally, CD4^+^ T cells have been regarded as providing help to CD8^+^ T cells in triggering an antitumor immune response. However, recent findings have indicated that CD4^+^ T cells may also possess direct antitumor capacity [24]. CD8^+^ T cells, also known as cytotoxic T lymphocytes, play a pivotal role in antitumor immunity, selectively detecting and eliminating tumor cells [25]. Moreover, in the Lewis lung carcinoma model, vaccination using an amino-rich FN nanoplatform loaded with tumor antigens and immunostimulatory CpG ODNs efficiently increases the CD3^+^CD8^+^, CD3^−^CD8^+^, and CD11b^+^CD80^+^ cell populations in the tumors, suggesting the alteration of the tumor microenvironment from cold to hot tumors. The increase in CD3^+^CD8^+^ T cell population in the tumor sites shows the same trend as that in the spleens, and the infiltration of cytotoxic CD3^+^CD8^+^ T cells in the tumor sites facilitates the detection and elimination of tumor cells [26,27]. Here, the increase in CD3^−^CD8^+^ cell population can be attributed to the increase in NK cells [23], because CpG ODNs readily trigger the immune activation of NK cells. The increase in CD11b^+^CD80^+^ cell population, representing M1 macrophages, provides a beneficial tumor microenvironment to suppress tumor growth [26].

## 5. Conclusions

In this study, a green one-step synthesis process was used to prepare an amino-rich metal–organic nanoplatform (FN). The synthesized FN nanoplatform can simultaneously and effectively load model tumor antigens (OVA)/autologous tumor antigens (dLLC) and immunostimulatory CpG ODNs with an unmodified PD backbone and a guanine quadruplex structure to obtain various cancer vaccines. An FN nanoplatform and immunostimulatory CpG ODNs generate synergistic effects to enhance the immunogenicity of different antigens and inhibit the growth of established and distant tumors in both the murine E.G7-OVA lymphoma model and the murine Lewis lung carcinoma model. In the E.G7-OVA lymphoma model, vaccination efficiently increases the CD4^+^, CD8^+^, and tetramer^+^CD8^+^ T cell populations in the spleens. In the Lewis lung carcinoma model, vaccination efficiently increases the CD3^+^CD4^+^ and CD3^+^CD8^+^ T cell populations in the spleens and CD3^+^CD8^+^, CD3^−^CD8^+^, and CD11b^+^CD80^+^ cell populations in the tumors, suggesting the alteration of the tumor microenvironment from cold to hot tumors.

## Figures and Tables

**Figure 1 vaccines-12-00649-f001:**
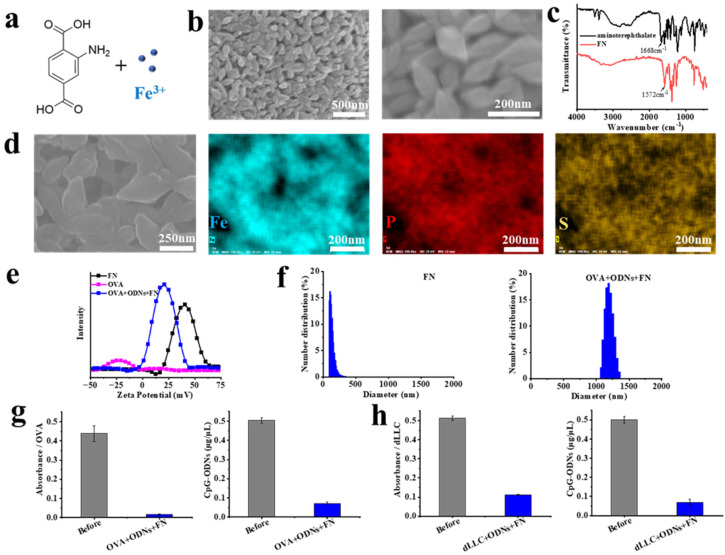
Physicochemical characterization of FN nanoplatform and cancer vaccines. (**a**) Schematic illustration of the synthesis of FN nanoplatform using 2-aminoterephthalic acid and Fe^3+^. (**b**) SEM images of FN nanoplatform. (**c**) FTIR spectra of reactant (2-aminoterephthalic acid) and reaction product (FN). (**d**) SEM image and EDX mapping analysis of cancer vaccines for E.G7-OVA lymphoma (OVA + ODNs + FN). Uniform distribution of Fe, P, and S elements suggests that model antigen OVA and CpG ODNs are homogeneously adsorbed into FN nanoplatform. (**e**) Zeta potentials of FN nanoplatform, model antigen OVA and cancer vaccines (OVA + ODNs + FN). (**f**) DLS analysis of FN nanoplatform and cancer vaccines (OVA + ODNs + FN). (**g**,**h**) Absorbance of proteins and concentration of CpG ODNs before and after loading into cancer vaccines for E.G7-OVA lymphoma (OVA + ODNs + FN) and cancer vaccines for Lewis lung carcinoma (dLLC + ODNs + FN).

**Figure 2 vaccines-12-00649-f002:**
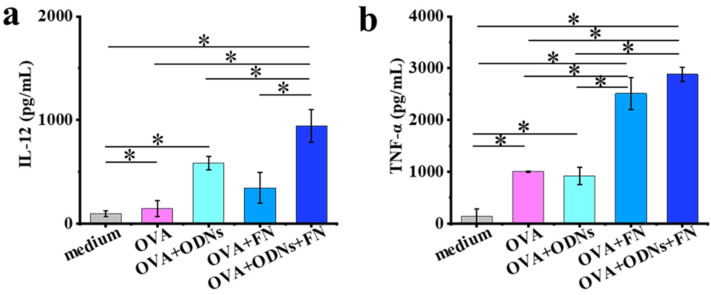
Quantitative analysis of cytokine contents after culture for 1 day (**a**) or 3 days (**b**). Data are presented as mean ± SD (n = 3). * *p* < 0.05.

**Figure 3 vaccines-12-00649-f003:**
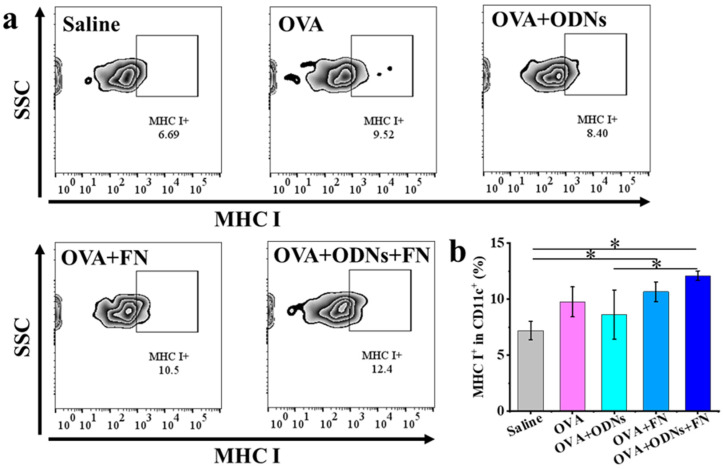
Representative flow cytometry plots (**a**) and quantitative analysis (**b**) of MHC I^+^ in CD11c^+^ cells population in lymph nodes of mice. Data are presented as mean ± SD (n = 3). * *p* < 0.05.

**Figure 4 vaccines-12-00649-f004:**
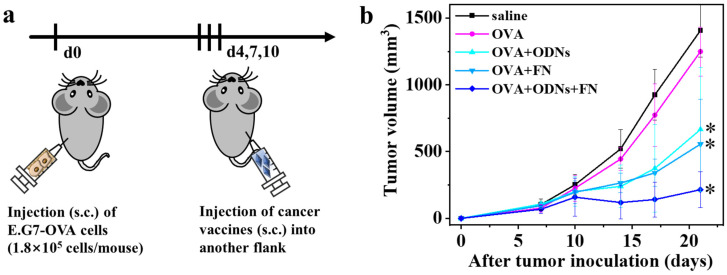
Therapeutic effects in E.G7-OVA lymphoma-bearing mouse model. (**a**) Schematic diagram of antitumor experimental process; (**b**) tumor growth curves of mice administered with different formulations. Data are presented as mean ± SD (n = 6). * *p* < 0.05.

**Figure 5 vaccines-12-00649-f005:**
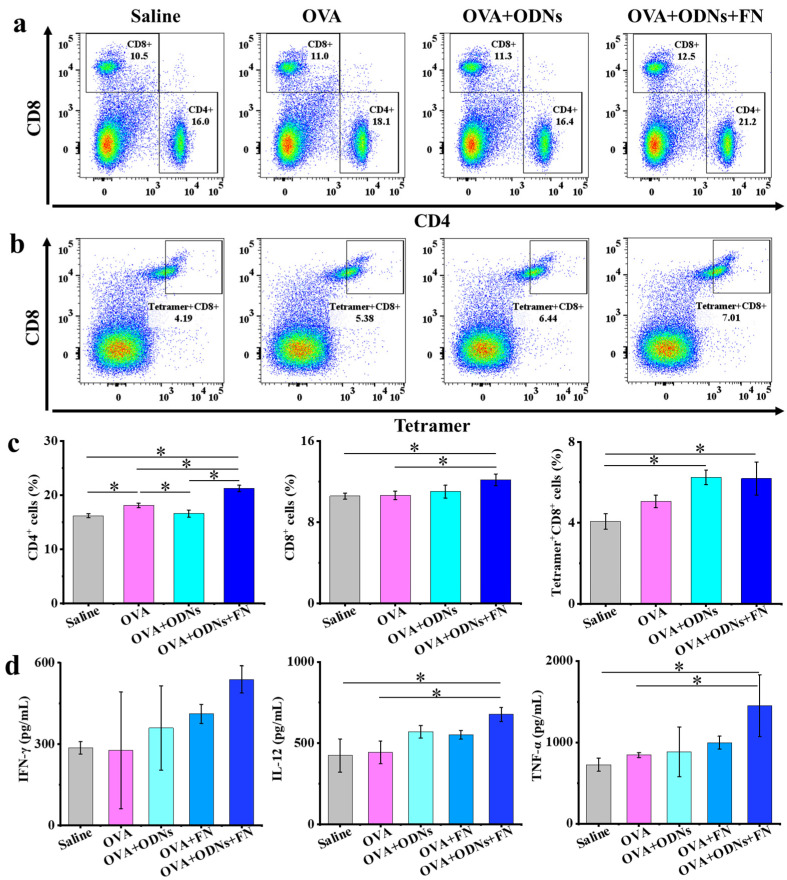
Antitumor mechanism analysis in therapeutic E.G7-OVA lymphoma mouse model. (**a**–**c**) Representative flow cytometry plots (**a**,**b**) and populations (**c**) of CD4^+^, CD8^+^, and tetramer^+^CD8^+^ T cells in spleen at the endpoint. (**d**) Quantitative analysis of cytokine content in spleen at the endpoint. Data are presented as mean ± SD (n = 3). * *p* < 0.05.

**Figure 6 vaccines-12-00649-f006:**
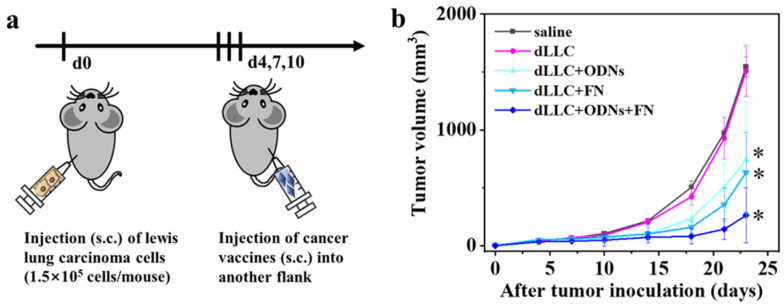
Therapeutic effects in Lewis lung carcinoma-bearing mouse model. (**a**) Schematic diagram of antitumor experimental process; (**b**) tumor growth curves of mice administered with different formulations. Data are presented as mean ± SD (n = 4). * *p* < 0.05.

**Figure 7 vaccines-12-00649-f007:**
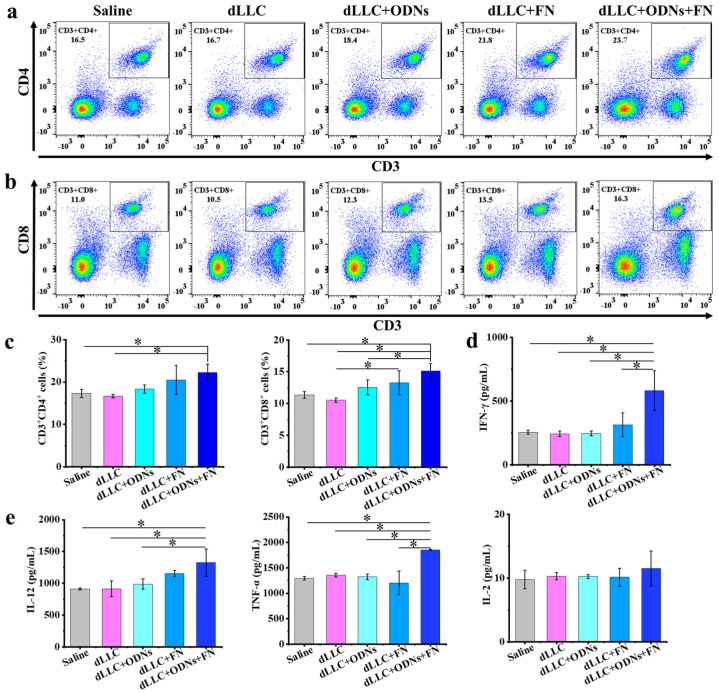
Antitumor mechanism analysis in therapeutic Lewis lung carcinoma mouse model. (**a**–**c**) Representative flow cytometry plots (**a**,**b**) and populations (**c**) of CD3^+^CD4^+^ and CD3^+^CD8^+^ in spleen at the endpoint. (**d**,**e**) Quantitative analysis of cytokine content in spleen at the endpoint. Data are presented as mean ± SD (n = 4). * *p* < 0.05.

**Figure 8 vaccines-12-00649-f008:**
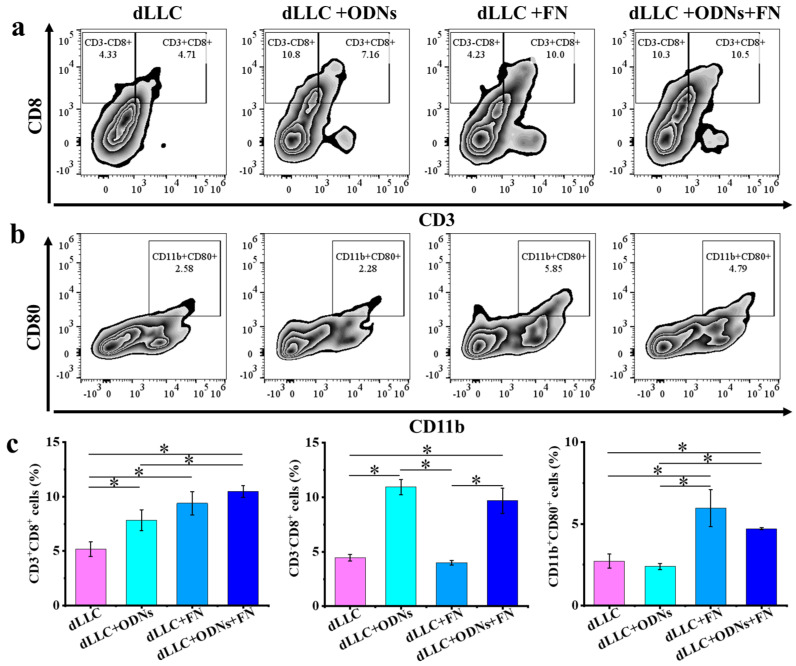
Antitumor mechanism analysis in therapeutic Lewis lung carcinoma mouse model. Representative flow cytometry plots (**a**,**b**) and populations (**c**) of CD3^+^CD8^+^, CD3^−^CD8^+^, and CD11b^+^CD80^+^ cells in tumor sites at the endpoint. Data are presented as mean ± SD (n = 4). * *p* < 0.05.

## Data Availability

The data supporting the findings of this study are available upon request from the corresponding authors.

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
