# Peer review of "Synergistic Effects of Metal–Organic Nanoplatform and Guanine Quadruplex-Based CpG Oligodeoxynucleotides in Therapeutic Cancer Vaccines with Different Tumor Antigens"

_vaccines, 2024, doi:10.3390/vaccines12060649_

Round 1

Reviewer 1 Report

Comments and Suggestions for Authors

1. The developed nanoparticles did not show satisfactory physicochemical characterization. Previously, the description of the methodology was inadequate and lacked scientific rigor. In the results section, the physicochemical characterization was inadequately described and not sufficiently justified.

2. The parameters used to analyze the immune response were also unsatisfactory.

3. For better understand the immune response, it is essential to analyze the cytokine network. Analysis of the cytokine network biomarkers would provide more precise insights.

4.  Although the authors describe the cellular mechanisms, the discussion of the research findings lacks scientific soundness.

Overall, the authors have attempted to describe the development of a cancer vaccine, but the study lacks scientific rigor. Therefore, the study should me rejected.

Reviewer 2 Report

Comments and Suggestions for Authors

This manuscript presents a novel nanodelivery system based on metal-organic nanoparticles for delivering immunostimulatory CpG ODNs, demonstrating enhanced efficacy in priming T cells, NK cells, and macrophages to inhibit the growth of Lewis Lung Cancer in vivo. Despite these promising findings, I have two main concerns:

1.     The manuscript extensively discusses the fabrication of nanoparticles and their therapeutic efficacy yet does not sufficiently address the underlying signaling mechanisms that prime the immune system. This aspect is crucial for understanding how the nanovaccine activates immune responses and should be explored to provide a comprehensive view of the nanovaccine's functionality.

2.     The conventional administration route for tumor vaccines is via subcutaneous injection at the neck site, yet this study employs an intratumoral injection. The rationale behind choosing this less typical method of administration over the conventional approach should be clarified to ensure the study's relevance and applicability to broader clinical practices.

Author Response

This manuscript presents a novel nanodelivery system based on metal-organic nanoparticles for delivering immunostimulatory CpG ODNs, demonstrating enhanced efficacy in priming T cells, NK cells, and macrophages to inhibit the growth of Lewis Lung Cancer in vivo. Despite these promising findings, I have two main concerns:

Response:

We thank the reviewer for the positive, helpful, and professional comments.

  1. The manuscript extensively discusses the fabrication of nanoparticles and their therapeutic efficacy yet does not sufficiently address the underlying signaling mechanisms that prime the immune system. This aspect is crucial for understanding how the nanovaccine activates immune responses and should be explored to provide a comprehensive view of the nanovaccine's functionality.

Response:

We are very grateful to the reviewers for their insightful suggestions.

It is generally considered that CpG-ODNs activate the TLR9 signaling pathway, and thus induce the release of cytokines, such as IL-12, IL-6, IL-1β, TNF-α and so on. In this study, the presence of FN nanoplatform can further increase the secretion of IL-12 and TNF-α. In fact, what the reviewers suggested is exactly what we plan to do in the future. It is difficult to conduct in-depth research on the signaling mechanism in a short period of time.

  1. The conventional administration route for tumor vaccines is via subcutaneous injection at the neck site, yet this study employs an intratumoral injection. The rationale behind choosing this less typical method of administration over the conventional approach should be clarified to ensure the study's relevance and applicability to broader clinical practices.

Response:

Perhaps we didn’t describe clearly in the manuscript causing the reviewer to misunderstand. In this study, we also employed the conventional administration route for vaccines (subcutaneous injection). We used subcutaneous injection at the back site, rather than subcutaneous injection at the neck site. We have further emphasized the subcutaneous injection of the as-prepared cancer vaccines in the revised manuscript.

Reviewer 3 Report

Comments and Suggestions for Authors

The article by Li et al. deals with the development of new formulation of therapeutic anti-cancer vaccine. This vaccine formulation consists of three components, namely, a tumor antigen, an amino- rich metal-organic nanoplatform as a carrier and an oligodeoxynucleotides containing unmethylated cytosine-phosphate-guanosine motifs as a TLR9-based immunity activator. These components are expected to assemble into combined nanoparticles. The authors describe in detail the physicochemical properties of this vaccine and provide a number of convincing evidence of its therapeutic efficacy using two models of mouse tumors in vivo, such as E.G7-OVA lymphoma and Lewis lung carcinoma.

The study was carried out at a high methodological level. Undoubtedly, the results provided have a high scientific novelty and significance.

I have a few minor remarks to be addressed, before the paper is published.

1. In the title, you emphasize the anti-cancer role of the two synergizing components,  CpG ODNs and  FN nanoplatform. This would be fully relevant if you have developed an anti-cancer drug based on the above-mentioned compounds. But, instead, in the manuscript you speak about the therapeutic vaccine (and submit it to the “Vaccines” journal). Certainly, the main component of any vaccine is the specific antigen, with all the importance of adjuvants and carriers. Therefore, I think, in the title the word “vaccine” should sound obligatory, and the tumor antigen as the supposed main anti-cancer player should be mentioned. In the current version, the title is rather misleading.

2. My second question is logically linked to the first one. From the whole bulk of the data presented it remains rather unclear, to what extent the observed effect of tumor treatment is accounted for the CpG ODNs – based innate immunity stimulation, and to what extent – the tumor antigen – based adaptive immunity? In other words, have you developed in fact a vaccine or a medicine? This point could be clarified if one of your controls in model tumor treatment was ODNs + FN, without any tumor antigen. Possibly, in this case you would see the same tumor degradation, with no role of the specific antigen…. I think, this point deserves to be discussed in the paper.

3. Line 169. “…tumor volume was calculated using the following formula: 1/2 × tumor length × tumor width2”. I’m sorry, I did not understand at all this formula. Most probably, in reality the shape of tumor is close to the spheroid. The volume of the sphere is calculated using the formula 4/3πR3, nothing similar to your formula. If you think the tumor is a parallelepiped, then you need to multiply its length by width by height. Again, nothing similar to what you wrote. If you think, the tumor’s width is equal to its height (what is far not self-obvious), you can use “tumor length × tumor width2”, but why are you dividing this number by two?? Please explain this.

4. Lines 175-177. “In addition, the spleen was treated with a tissue protein extraction reagent (Thermo Fisher Scientific, USA) and cytokine contents were tested by ELISA kits (BD Biosciences)”. The “Results” section does not contain these data. So, please, either show these results or remove this sentence from the “Methods”.

5. When analyzing E.G7-OVA lymphoma mouse model and Lewis lung carcinoma mouse model, you used different sets of markers and finally collected different sets of data. Particularly, in splenocyte study you applied anti-CD3 antibody for the LLC and did not apply it for the E.G7-OVA lymphoma. Also, you studied the tumor itself for a number of markers (what is very important!) in the case of LLC, and did not do that in the case of lymphoma. I think, any interested reader will ask the question “why?” here. Perhaps, the text of the manuscript needs some words to explain that.   

Author Response

The article by Li et al. deals with the development of new formulation of therapeutic anti-cancer vaccine. This vaccine formulation consists of three components, namely, a tumor antigen, an amino- rich metal-organic nanoplatform as a carrier and an oligodeoxynucleotides containing unmethylated cytosine-phosphate-guanosine motifs as a TLR9-based immunity activator. These components are expected to assemble into combined nanoparticles. The authors describe in detail the physicochemical properties of this vaccine and provide a number of convincing evidence of its therapeutic efficacy using two models of mouse tumors in vivo, such as E.G7-OVA lymphoma and Lewis lung carcinoma.

The study was carried out at a high methodological level. Undoubtedly, the results provided have a high scientific novelty and significance.

I have a few minor remarks to be addressed, before the paper is published.

Response:

We thank the reviewer for the positive, insightful, and professional comments. We have carefully revised the manuscript according to the reviewer’s suggestions.

  1. In the title, you emphasize the anti-cancer role of the two synergizing components, CpG ODNs and FN nanoplatform. This would be fully relevant if you have developed an anti-cancer drug based on the above-mentioned compounds. But, instead, in the manuscript you speak about the therapeutic vaccine (and submit it to the “Vaccines” journal). Certainly, the main component of any vaccine is the specific antigen, with all the importance of adjuvants and carriers. Therefore, I think, in the title the word “vaccine” should sound obligatory, and the tumor antigen as the supposed main anti-cancer player should be mentioned. In the current version, the title is rather misleading.

Response:

We totally agree with the reviewer. We have revised the title of the manuscript to “Synergistic effects of metal-organic nanoplatform and guanine quadruplex-based CpG oligodeoxynuceotides in therapeutic cancer vaccines with different tumor antigens”

  1. My second question is logically linked to the first one. From the whole bulk of the data presented it remains rather unclear, to what extent the observed effect of tumor treatment is accounted for the CpG ODNs – based innate immunity stimulation, and to what extent – the tumor antigen – based adaptive immunity? In other words, have you developed in fact a vaccine or a medicine? This point could be clarified if one of your controls in model tumor treatment was ODNs + FN, without any tumor antigen. Possibly, in this case you would see the same tumor degradation, with no role of the specific antigen…. I think, this point deserves to be discussed in the paper.

Response:

We are very grateful for the reviewer's nice comments. We have added the discussion about this point in the revised manuscript (Line 413-420)

“In addition, it should be mentioned that FN nanoplatform loaded with CpG ODNs is designed to be the adjuvants for therapeutic cancer vaccines in this study. Therefore, subcutaneous administration of the as-prepared cancer vaccines was employed to trigger systemic antitumor immune response and the inhibition effect on the distant tumors were investigated. In fact, intratumoral or intravenous administration of FN nanoplatform loaded with CpG ODNs as a medicine is also promising for cancer treatment and the relevant work using different antitumor experimental process will be considered in the future.”

  1. Line 169. “…tumor volume was calculated using the following formula: 1/2 × tumor length × tumor width2”. I’m sorry, I did not understand at all this formula. Most probably, in reality the shape of tumor is close to the spheroid. The volume of the sphere is calculated using the formula 4/3πR3, nothing similar to your formula. If you think the tumor is a parallelepiped, then you need to multiply its length by width by height. Again, nothing similar to what you wrote. If you think, the tumor’s width is equal to its height (what is far not self-obvious), you can use “tumor length × tumor width2”, but why are you dividing this number by two?? Please explain this.

Response:

The reviewer mentioned a very important and interesting question.

The tumor is assumed to be an ellipsoid. Ellipsoid volume is calculated based on the formula: width x height x length x π / 6. However, because height is difficult to measure, the following modified formular is widely used when calipers are involved: 1/2 × tumor length × tumor width2.

[ref] Subcutaneous tumor volume measurement in the awake, manually restrained mouse using MRI, https://doi.org/10.1002/jmri.23829

  1. Lines 175-177. “In addition, the spleen was treated with a tissue protein extraction reagent (Thermo Fisher Scientific, USA) and cytokine contents were tested by ELISA kits (BD Biosciences)”. The “Results” section does not contain these data. So, please, either show these results or remove this sentence from the “Methods”.

Response:

We are very grateful to the reviewer for pointing out this problem. We apologize for our negligence. We have supplemented some cytokines data in the revised manuscript, such as (new Figure 5d and new Figure 7d, e)

  1. When analyzing E.G7-OVA lymphoma mouse model and Lewis lung carcinoma mouse model, you used different sets of markers and finally collected different sets of data. Particularly, in splenocyte study you applied anti-CD3 antibody for the LLC and did not apply it for the E.G7-OVA lymphoma. Also, you studied the tumor itself for a number of markers (what is very important!) in the case of LLC, and did not do that in the case of lymphoma. I think, any interested reader will ask the question “why?” here. Perhaps, the text of the manuscript needs some words to explain that.

Response:

We are very grateful for the reviewer's helpful suggestions. We have added the explanation in the revised manuscript.

Line 421-428,

“For mouse models and different groups used in this study, the CD4+ and CD3+CD4+ cell populations in spleen are almost the same, and the CD8+ and CD3+CD8+ cell populations in spleen are almost the same, as shown in Figure 7. But for tumor tissue, the situation is much more complicated. The CD3+CD8+ T cell populations in tumors represent the infiltration of cytotoxic T cells in the tumor sites. While the CD3-CD8+ cell populations can represent NK cells in the tumor sites in this study. Therefore, we stained the cells derived from spleens and tumors in Lewis lung carcinoma - bearing mouse model using anti-CD3 antibody to clarify this difference in splenocytes and tumor cells.”

In vivo anti-tumor experiment using therapeutic E.G7-OVA lymphoma mouse model was performed one year earlier than that using therapeutic Lewis lung carcinoma mouse model. Frankly speaking, in vivo anti-tumor experiment in the therapeutic E.G7-OVA lymphoma mouse model is a preliminary screening experiment. The newly purchased tumor tissue digestion reagents and some staining reagents had not arrived yet, so the tumor tissue was not evaluated in this experiment.

In vivo anti-tumor experiment using therapeutic Lewis lung carcinoma mouse model was designed and performed based on the experimental results of that using therapeutic E.G7-OVA lymphoma mouse model. We hoped to study the anti-tumor mechanism more deeply and the immune responses in tumor tissue were also analyzed.

Although we once considered using the EG7 model to further study the immune response in the tumor site, we gave up the idea in the end due to the 3R principle of animal experiments.

Round 2

Reviewer 1 Report

Comments and Suggestions for Authors

-

Author Response

We thank the reviewer for the positive comments.

Reviewer 2 Report

Comments and Suggestions for Authors

I do not have further comments this time.

Author Response

(The authors gave the same response as above.)

Reviewer 3 Report

Comments and Suggestions for Authors

Dear authors,

you performed a detailed and thorough job on the paper improvement in line with my comments. Now the manuscript is in exellent shape and can be published. I only have one small request left. Could you please transfer your explanation of the tumor volume formula from the rebuttal letter to the "Methods", with the reference which you gave? Otherwise, readers would experience difficulties in its understanding, similar to how I did. 

Author Response

We thank the reviewer for the positive comments. We have revised the manuscript as suggested.